# "All of the things to everyone everywhere": A mixed methods analysis of community perspectives on equitable access to monoclonal antibody treatment for COVID-19

Bethany M. Kwan[1,2,3,4]*, Chelsea Sobczak[2], Carol Gorman[4], Samantha Roberts[5], Vanessa Owen[2], Matthew K. Wynia[3,6,7], Adit A. Ginde[1,3], Griselda Pena-Jackson[2,8], Owen Ziegler[9], Lisa Ross DeCamp[4,10,11]

1 Department of Emergency Medicine, University of Colorado School of Medicine, Aurora, Colorado, United States of America, 2 Department of Family Medicine, University of Colorado School of Medicine, Aurora, Colorado, United States of America, 3 Colorado Clinical & Translational Sciences Institute, University of Colorado Anschutz Medical Campus, Aurora, Colorado, United States of America, 4 Adult & Child Center for Outcomes Research & Delivery Science, University of Colorado Anschutz Medical Campus, Aurora, Colorado, United States of America, 5 Department of Biostatistics and Informatics, Colorado School of Public Health, Aurora, CO, United States of America, 6 Center for Bioethics and Humanities, University of Colorado Anschutz Medical Campus, Aurora, Colorado, United States of America, 7 Division of General Internal Medicine, Department of Medicine, University of Colorado School of Medicine, Aurora, Colorado, United States of America, 8 2040 Partners for Health, Aurora, Colorado, United States of America, 9 Z Cultural Services, Denver, Colorado, United States of America, 10 Department of Pediatrics, University of Colorado School of Medicine, Aurora, Colorado, United States of America, 11 Latino Research and Policy Center, Colorado School of Public Health, Aurora, Colorado, United States of America

* bethany.kwan@cuanschutz.edu

**Data Availability Statement:** Data cannot be shared publicly because participants did not provide consent. Data are available from the

## Abstract

### Background

Neutralizing monoclonal antibody (mAb) treatment for COVID-19 prevents hospitalization and death but is underused, especially in racial/ethnic minority and rural populations. Reasons for underuse and inequity may include community member lack of awareness or healthcare access barriers, among others. This study assessed mAbs community awareness and opportunities for improving equitable mAb access.

### Methods

A concurrent mixed methods study including surveys and focus groups with adults with high-risk conditions or their proxy decision-makers. Surveys and focus group guides addressed diffusion of innovation theory factors. Descriptive statistics and Fisher's exact method was used to report and compare survey findings by race and ethnicity. Rapid qualitative methods were used for focus group analysis.

### Results

Surveys from 515 individuals (460 English, 54 Spanish, 1 Amharic), and 8 focus groups (6 English, 2 Spanish) with 69 participants, completed June 2021 to January 2022. Most

University of Colorado Office of the Chief Research Informatics Officer (melissa.haendel@cuanschutz.edu) for researchers who meet the criteria for access to confidential data. Please reference "Data availability request - Bethany Kwan - COMIRB #21-2747" in your request.

**Funding:** This publication was supported by grants from the National Institutes of Health (NIH; https://nih.gov/)/National Center for Advancing Translational Sciences (NCATS; https://ncats.nih.gov/) Colorado CTSA Grant Number UL1-TR002535-03 (PI: Ronald Sokol) and 3UL1TR002535-03S3 (MPI: Ronald Sokol and Adit Ginde). Its contents are the authors' sole responsibility and do not necessarily represent official NIH views. The funders had no role in the study design, data collection and analysis, decision to publish, or preparation of the manuscript.

**Competing interests:** "One author reports receiving grants from NIH, CDC, DoD, and pharmaceutical companies outside the submitted work. The other authors have declared that no competing interests exist."

survey respondents (75%) had heard little or nothing about mAbs, but 95% would consider getting mAb treatment. Hispanic/Latino and Non-Hispanic People of Color (POC) reported less awareness, greater concern about intravenous infusions, and less trust in mAb safety and effectiveness than White, Non-Hispanic respondents. Focus group themes included little awareness but high interest in mAb treatment and concerns about cost and access barriers such as lacking established sources of care and travel from rural communities. Focus groups revealed preferences for broad-reaching but tailored messaging strategies using multiple media and trusted community leaders.

## Conclusions

Despite unfamiliarity with mAb treatment, most respondents were open to receiving mAbs or recommending mAbs to others. While mAb messaging should have broad reach "to everyone everywhere," racial and geographic disparities in awareness and trust about mAbs underscore need for tailored messaging to promote equitable access. Care processes should address patient-level barriers like transportation, insurance, or primary care access. COVID-19 treatment dissemination strategies should promote health equity.

## Introduction

For the first year of the COVID-19 global pandemic, little could be done to effectively prevent or treat the disease beyond behavioral measures (e.g., social distancing) and symptom management. The introduction of neutralizing monoclonal antibody (mAb) treatments represented the first significant evidence-based advancement in outpatient treatment for COVID-19. The first mAb for COVID-19 (bamlanivimab) was authorized for emergency use in the United States in November 2020 to treat symptomatic COVID-19 among high-risk outpatient populations [1–3]. Additional mAbs were introduced thereafter, and availability and use fluctuated as different mAbs were determined to be more or less effective for different variants [1]. However, despite evidence of effectiveness, a strong recommendation from the National Institutes of Health (NIH), and medication provided free of cost to the patient, widespread use lagged availability and disparities in access and use were observed [4, 5]. Even now, as COVID-19 is increasingly seen as an endemic condition we must learn to manage [6], there remains a need to ensure timely access to treatment, particularly among racial/ethnic minorities with a disproportionate burden of disease [4, 5]. This paper presents a mixed methods analysis of community perspectives on equitable access to mAbs.

As with most health care innovations, there are multiple potential factors at the level of individual patients, families, communities, health care and public health systems, and the media that likely influenced uptake of mAbs for treatment of COVID-19 [7]. Presumably central is general community and health care provider awareness of the existence of evidence-based treatment for COVID-19 among those not yet sick enough to be hospitalized. Implementation factors for accessing care are likely also important. Originally, eligibility was restricted to adults and adolescents with a complex set of criteria indicating a high-risk of poor outcomes for COVID-19 [8]. Initially, treatment required a one-hour intravenous (IV) infusion delivered in a clinical setting with personnel qualified to monitor for adverse reactions [9]. The complex eligibility criteria and IV delivery of treatment requiring care in a well-staffed clinical facility may have contributed to slow uptake and demographic disparities in use [10].

People with symptoms of COVID-19 must be tested, diagnosed, receive a referral for treatment from a medical provider, secure and attend an appointment all within 10 days of first experiencing symptoms and without worsening to the point of requiring hospitalization to receive mAbs. These steps require timely action on the part of people experiencing symptoms of COVID-19 to get tested as well as knowledgeable, prepared health care providers, willing patients, and systems set up to deliver treatment efficiently. Over the course of 2021, eligibility criteria were expanded and simplified, an option for subcutaneous injection was introduced [11], and several states implemented policies designed to facilitate access (e.g., standing orders, self-referral, mobile treatment sites) [12–16]. For instance, the state of Colorado set up a secure web-based system to enable community health care providers to refer patients for mAb treatment at one of the multiple infusion sites across the state. Despite efforts to improve access and streamline processes, the multi-step process for accessing treatment–including requiring assessing of eligibility and a referral from a medical provider–may be a deterrent for patients and providers alike and a potential source of disparities in access and use of mAb treatment– especially in rural areas and populations with lower rates of insurance coverage [17].

A large-scale analysis of racial and ethnic disparities in access to mAb treatment for COVID-19 using patient electronic health record data from 41 U.S. health care systems was conducted by the Patient-Centered Outcomes Research Network (PCORNet) in 2021 [5]. Their analysis found that less than 2% of over 700,000 eligible mAb patients between November 2020 (when mAbs were authorized) and August 2021 received the treatment. Monthly mAb treatment rates increased over time, from less than 1% in November 2020 to approximately 6% in August 2021. Of those who received mAbs, racial/ethnic minority groups were underrepresented, with disparate access for Black, Asian, and Hispanic patients [5].

Thus, concerns about the inequitable access to and use of mAbs among marginalized patients with COVID-19 appear legitimate and observable nationally. To address these disparities in COVID-19 outcomes, understanding the factors influencing inequitable access to and treatment use is critical. For instance, it is unclear to what extent mAb treatment access disparities are due to community-level knowledge, cultural, or psychosocial factors (e.g., lack of awareness, trust in treatment) versus logistical factors (e.g., access to care, cost). A public health perspective on health disparities considers both individual-level factors such as health beliefs and system-level factors such as strategies to ensure the availability of care [18]. This study assessed community awareness and opportunities for improving equitable access to and use of mAbs in Colorado. We used concurrent quantitative and qualitative methods (surveys and focus groups) to understand the perspectives of community members in Colorado who might be eligible to receive mAb treatment if diagnosed with COVID-19 or who were proxy decision makers for those who might qualify for mAb treatment.

## Methods

### Study design and setting

This study used a concurrent mixed methods design to assess community awareness, trust, willingness to receive treatment, and opportunities for community messaging about mAb treatment for COVID-19 in the U.S. state of Colorado. A concurrent mixed methods approach [19] uses triangulation of quantitative and qualitative findings to gain a deeper understanding and identify consistent/inconsistent themes among surveys and focus groups. The study was conducted from June 2021-January 2022. Rogers' diffusion of innovations theory was used to guide the design of a community survey and a focus group guide [20]. A parallel component of the project assessed health care provider perspectives, and informed a provider-level dissemination strategy, reported elsewhere [21]. This research was approved as exempt human

subjects research by the Colorado Multiple Institutional Review Board (COMIRB Protocol #21–2747). The research was conducted in accordance with universal ethical principles. All participants provided written informed consent using an information consent process embedded in the survey and focus group procedures.

Results were shared with the project's multi-stakeholder advisory panel (SAP) and informed the co-design of community-level mAb treatment messaging, packaging, and distribution of materials to community members and health care providers. The SAP was comprised of 25 individuals including 12 community members, 3 healthcare providers, 2 public health department representatives, and 8 regional health connectors (RHCs). SAP members were recruited through professional contacts and existing relationships with community organizations and practice-based research networks. The RHCs are a community-based workforce in Colorado based in health organizations across the state with the goal of identifying and addressing health issues with their regions. Findings also were shared with Colorado policymakers to identify strategies for improving equitable access to and use of mAb treatment.

## Participants and recruitment

Eligibility criteria for surveys included adults at least 18 years of age living in Colorado who self-identified as high risk for poor outcomes from COVID-19 or as a proxy decision maker for someone at high risk. We aimed to recruit 450 people to participate in surveys and between 60 and 80 people to participate in 8 focus groups (6 English, 2 Spanish, 8–10 people each). The original 450 survey sample size estimation was based on plans to assess differences in willingness to receive mAb treatment across a range of groups (e.g., geographic regions, demographic groups). At this sample size, we estimated 80% power to detect a difference of 18% among 9 groups of 50 people. For this analysis (differences among 3 racial/ethnic groups), we estimated 80% power to detect a difference of 15% among groups of at least 50 people per group. To address geographic and racial/ethnic disparities in access to care and COVID-19 outcomes, we sought to overrepresent respondents who identified as Hispanic/Latino relative to state demographics and those from rural areas. The survey was made available in 7 languages, including English, Spanish, French, Amharic, Russian, Korean, and Vietnamese, based on recommendations from the project's stakeholder advisory panel (SAP). A commercial transcription and translation company, Landmark Associates, performed survey translation and reviewed to confirm wording and meaning in the target language prior to use. Survey recruitment was facilitated through the collaboration of university and community team members, including: the SAP members, the University of Colorado's Practice Innovation Program network, PEACHnet (a geographically based community and practice-based research network that collaborates with partners across the western slope of Colorado) [22], local public health agencies, and local community organizations. Survey recruitment flyers were distributed via email or in-person to over 100 community organizations, public entities such as libraries and health departments, and primary care practices or providers. Flyers were co-designed by SAP members and were available in Spanish, English, and French. The mAb Colorado team shared the survey link on the project website and posted it on the project's media platforms (Facebook and Twitter); these posts were also shared by various health departments and community members on their social media.

Potential focus group participants were a convenience sample identified from 373 survey respondents who had indicated they agreed to further contact regarding related research studies during survey completion. Survey participants who agreed to further contact were stratified by the language of survey completion (English, Spanish). English-language survey participants were then further stratified using the ZIP code they provided on the survey as living in an

urban/suburban or rural region of the state. Based on the preferred method of contact listed in the survey, participants in three strata (English-Urban/Suburban, English-Rural, Spanish) were contacted to participate in a focus group. Potential participants were emailed in batches of 20 or texted in batches of 10. Potential participants were contacted up 3 times before no further contact was made. Participants who responded to participation requests selected an available focus group time. Of those willing to be contacted, 258 people (69.2%) did not respond to contact or declined participation in focus groups. This method of focus group recruitment did not yield a sufficient number of Spanish-speakers to conduct a planned second Spanish-language focus group. We worked with a community-based organization to identify additional participants from the Hispanic/Latino community. Staff from the community-based organization recruited additional participants from clients served by their organization to complete the survey and participate in a focus group.

## Survey measures and data collection

The survey included 4 screening items, 18 items assessing demographics and health status, 15 items assessing awareness and experiences with COVID-19 tests, vaccines, and treatments, and 23 items about trust and willingness to receive mAb treatment (16 items about receiving treatment oneself, and 7 items about accepting/recommending treatment as a proxy decision maker). The survey items were developed specifically for this study. We tested an original survey draft for general flow, length, and understandability with 6 community members and subsequently revised the survey. See S1 Appendix for the final survey. The survey was administered electronically using REDCap electronic data capture tools hosted at the University of Colorado [23]. Participants with valid responses received a $25 e-gift card.

To identify potentially invalid responses [24], which were common due to the public availability of the survey link, we reviewed responses for missing answers and logic checks. Survey responses were downloaded weekly and reviewed in Excel, using conditional formatting to identify red flags such as missing or duplicate addresses, repeat free text responses in large clusters, duplicate names, and ZIP codes outside of Colorado. We also scanned responses for patterns in names, and email addresses. Logic checks were also done within survey items (i.e., if a respondent said they had never been tested for COVID-19 but had received a positive test for COVID-19; or if a respondent said they had never had COVID-19 but had received monoclonal antibody treatment for COVID-19). Finally, we conducted an internet search for each address to confirm it was a legitimate residential address in Colorado. We confirmed residence at that address through internet searches for any responses that raised the above red flags.

## Focus group data collection

Focus group discussion guide topics include knowledge about mAbs, questions or concerns about mAb treatment, whether they would consider this treatment, perceptions on the process of obtaining treatment, and potential methods to inform members of their community about mAb treatment. The focus group guide can be found in S2 Appendix. All focus groups were conducted virtually using the Zoom virtual meeting platform. Each focus group was moderated by one of two study investigators with more than 10 years of training and experience in qualitative research and a research staff member with 7 years of experience in qualitative research (CG). CG completed focus group recruitment. Both investigators identify as a woman (BK (PhD) and LRD (MD; bilingual in English and Spanish)). The research staff member is a woman who is bilingual English/Spanish. Focus groups lasted between 60–90 minutes, and participants were remunerated $100 for their participation. No other people besides participants and moderators attended focus groups. The focus groups began with brief introductions

by the moderators and participants who did not know each other before the focus group. During introductions, moderators and participants shared their pronouns, location, hobbies/interests and responded to an icebreaker question (e.g., What is your favorite movie). After introductions, there was an initial inquiry into mAbs knowledge. Then participants were shown a brief Combat Covid video [25] about mAbs for COVID-19, and the moderator reviewed a flyer created by the mAb Colorado team that provided information on the process of obtaining mAbs treatment in Colorado (both videos and flyers are available in English and Spanish) [26]. These resources were used during focus groups based on the survey findings of limited knowledge of mAbs so that focus group participants could provide their views with more baseline knowledge about the treatment and how to obtain it. Focus groups were recorded using the Zoom platform and transcribed/translated by a commercial transcription company. Any identifying information was removed from transcripts. Transcripts were not returned to participants for comment or correction.

## Quantitative analysis

We used chi-square and Fisher's exact tests for categorical variables and t-tests for continuous variables to compare demographic information, awareness and experience with COVID-19 tests, vaccines and treatments, and trust in monoclonal antibody treatment between participants who self-reported as White Non-Hispanic, Hispanic/Latino (any race), or other Non-Hispanic People of Color (POC). Using chi-square and Fisher's exact tests for categorical variables and t-tests for continuous variables, we compared demographic information between participants who completed the focus group to those that did not. We performed sub-analyses to compare responses among Hispanic participants by the language of survey completion (Spanish vs. English) to assess whether there were additional concerns among those with a language barrier. We created a bar plot to show all respondents' awareness of COVID-19 tests, vaccines, and treatments.

## Qualitative analysis

Focus group transcripts were analyzed using rapid qualitative methods [27] based on predefined areas of interest to inform the dissemination of mAbs by one investigator who moderated the focus groups (LRD) and the research staff member who co-moderated (CG). Field notes made during the focus groups were reviewed before analysis in conjunction with a review of transcripts to confirm data saturation. However, they were not analyzed or used to draw conclusions as it was determined they offered no additional relevant information to inform analytic conclusions. Transcripts were summarized in a table in Microsoft Word by research staff members with rows corresponding to focus group discussion topics (e.g., Baseline knowledge about monoclonal antibodies, Willingness to receive mAb treatment). Differences across summaries were resolved through consensus. Using matrix analysis techniques [28], summaries were categorized into English-Urban/Suburban, English-Rural, and Spanish. There were insufficient numbers of Spanish-speaking focus group participants to permit similar distinctions for geographic location. The two research team members identified and grouped key themes from summaries. They then compared themes across the focus group types using a saturation grid [29], which visually aligns support for themes across focus group types to cross-check and confirm evidence of themes against the focus group data. We did not complete member-checking with participants, but findings were shared with the project's community stakeholder advisory panel, who endorsed findings that reflected community member experiences and viewpoints.

## Results

### Sample characteristics

A total of 515 participants completed the survey; 54 completed it in Spanish, 460 in English, and one in Amharic. Sixty-nine survey respondents participated in a subsequent focus group, with a total of 6 focus groups conducted. Table 1 shows sample characteristics for survey and focus group participants. Survey respondents were predominantly women, employed, with a wide range of educational backgrounds, and from various racial/ethnic groups. Racial/ethnic minorities were overrepresented relative to the Colorado population, with a greater proportion of Hispanic respondents than US census data for Colorado (27.4% survey vs 21.8% census), more Black/African American (6.8% survey vs 4.6% census), more mixed race (5.4% survey vs 3.1% census), and fewer White Non-Hispanic (54.9% survey vs 67.7% census). About 89.5% of survey respondents indicated that they lived in a metropolitan area, while 8.5% resided in a

**Table 1. Participant characteristics.**

| | Overall (n = 515) | Non-Focus Group Participant (n = 445) | Focus Group Participant (n = 69) | p** |
|---|---|---|---|---|
| Age in Years: Mean (SD) | 44.4 (13.9) | 43.9 (13.9) | 47.6 (13.7) | .04 |
| Gender: N (%) | | | | .96** |
| Man | 120 (23.4) | 105 (23.6) | 15 (22.1) | |
| Woman | 383 (74.7) | 331 (74.4) | 52 (76.5) | |
| Non-binary | 9 (1.8) | 8 (18) | 1 (1.5) | |
| Prefer not to say | 1 (0.2) | 1 (0.2) | 0 (0.0) | |
| Race/Ethnicity: N (%) | | | | .45** |
| Black/African American | 36 (7.0) | 32 (7.2) | 4 (5.8) | |
| White/Caucasian | 282 (55.0) | 248 (55.9) | 34 (49.3) | |
| Hispanic or Latino/a (any race) | 141 (27.5) | 115 (25.9) | 26 (37.7) | |
| Asian | 12 (2.3) | 12 (2.7) | 0 (0.0) | |
| Native Hawaiian/Pacific Islander | 1 (0.2) | 1 (0.2) | 0 (0.0) | |
| Native American/Alaska Native | 9 (1.8) | 7 (1.6) | 2 (2.9) | |
| Other | 4 (0.8) | 4 (0.9) | 0 (0.0) | |
| More Than One Race | 28 (5.5) | 25 (5.6) | 3 (4.3) | |
| Employment Status: N (%) | | | | .21 |
| Employed | 391 (75.9) | 343 (76.9) | 48 (69.6) | |
| Not Employed | 99 (19.2) | 84 (18.8) | 15 (21.7) | |
| Other/Prefer not to answer/No response | 25 (4.9) | 19 (4.3) | 6 (8.7) | |
| Education: N (%) | | | | .01** |
| High School/Some College | 170 (33.0) | 151 (33.9) | 19 (27.5) | |
| College Degree | 203 (39.4) | 183 (41.0) | 20 (29.0) | |
| Post Graduate Degree | 131 (25.4) | 104 (23.3) | 27 (39.1) | |
| Prefer not to say/No Response | 11 (2.1) | 8 (1.8) | 3 (4.3) | |
| History of COVID-19: N (%) Confirmed/Suspected COVID-19 | 188 (36.9) | 166 (37.7) | 22 (31.9) | .423 |
| Hospitalized for COVID-19: N (% of those with COVID-19) | 9 (1.7) | – | – | |
| Any High-risk Health Conditions:* N (%) Yes | 319 (61.9) | 271 (60.9) | 48 (69.6) | .20 |
| Received COVID-19 Vaccine (at least one dose): N (%) Yes | 447 (86.8) | 387 (86.8) | 60 (87.0) | 1.00 |

*High-risk health conditions, by self-report: Diabetes, Heart Disease, Kidney Disease, Hypertension, Obesity, Respiratory, Medication to suppress immune system, health conditions associated with compromised immune system

**Indicates results of an exact test

micropolitan, small town or rural area; respondents spanned 30 of the 64 counties in the state. About 60% of respondents indicated they had at least one condition that put them at high risk for poor outcomes for COVID-19 (and thus would be eligible for mAbs). About a third of participants reported they had confirmed or suspected COVID-19, and 4.8% of those indicated they were hospitalized for COVID-19. Compared with survey participants that did not participate in a focus group, focus group participants were somewhat older and had higher levels of education. There were no other statistically significant differences in demographics between focus group and non-focus group participants.

## Quantitative results

As shown in Table 2 and Fig 1, respondents overall indicated substantially more awareness of tests for COVID-19 (91.2% moderate or a lot of awareness) and vaccines to prevent COVID-19 (93.9% moderate or a lot of awareness) than for mAbs (25.8% moderate or a lot of awareness) or other medications to treat COVID-19 (21.3% moderate or a lot of awareness). There were significant racial/ethnic differences in awareness of vaccines and mAbs for COVID-19, with White Non-Hispanic respondents consistently reporting greater awareness than Hispanic/Latino and Non-Hispanic POC respondents. About 75% of respondents had been tested for COVID-19, and ~87% had been vaccinated (slightly higher than Colorado's vaccination rate at the time: as of 10/1/2021, 76.7% had received 1+ dose, and 70.3% were considered "up

**Table 2. Awareness of and experience with tests, vaccines, and treatments for COVID-19 by Race/Ethnicity.**

|  | Overall | Hispanic/ Latino/a | White Non-Hispanic | Non-Hispanic POC | p |
|---|---|---|---|---|---|
| n | 515 | 141 | 282 | 90 | |
| **Awareness of COVID-19 Tests: N (%)** | | | | | .069* |
| Nothing at all | 10 (2.0) | 7 (5.0) | 2 (0.7) | 1 (1.1) | |
| A little | 35 (6.8) | 11 (7.9) | 17 (6.0) | 7 (7.8) | |
| A moderate amount | 133 (26.0) | 38 (27.3) | 67 (23.8) | 27 (30.0) | |
| A lot | 334 (65.2) | 83 (59.7) | 195 (69.4) | 55 (61.1) | |
| **Awareness of COVID-19 Vaccines:: N (%)** | | | | | .023* |
| Nothing at all | 6 (1.2) | 4 (2.8) | 1 (0.4) | 1 (1.1) | |
| A little | 25 (4.9) | 9 (6.4) | 9 (3.2) | 7 (7.9) | |
| A moderate amount | 95 (18.6) | 34 (24.1) | 47 (16.8) | 14 (15.7) | |
| A lot | 385 (75.3) | 94 (66.7) | 222 (79.6) | 67 (75.3) | |
| **Awareness of mAbs: N (%)** | | | | | < .001 |
| Nothing at all | 186 (36.3) | 54 (38.6) | 81 (28.8) | 50 (55.6) | |
| A little | 195 (38.0) | 52 (37.1) | 120 (42.7) | 22 (24.4) | |
| A moderate amount | 95 (18.5) | 24 (17.1) | 61 (21.7) | 10 (11.1) | |
| A lot | 37 (7.2) | 10 (7.1) | 19 (6.8) | 8 (8.9) | |
| **Awareness of Other Medications for COVID-19: N (%)** | | | | | .28 |
| Nothing at all | 174 (34.2) | 53 (38.7) | 84 (30.0) | 36 (40.0) | |
| A little | 226 (44.4) | 53 (38.7) | 136 (48.6) | 37 (41.1) | |
| A moderate amount | 73 (14.3) | 21 (15.3) | 42 (15.0) | 9 (10.0) | |
| A lot | 36 (7.1) | 10 (7.3) | 18 (6.4) | 8 (8.9) | |
| **Received Medical Tests for COVID-19: N (%) Yes** | 388 (75.3) | 110 (78.0) | 209 (74.1) | 67 (74.4) | .67 |
| **Received Vaccines to prevent COVID-19: N (%) Yes** | 447 (86.8) | 119 (84.4) | 251 (89.0) | 76 (84.4) | .31 |
| **Received Monoclonal antibody medications to treat COVID-19: N (%) Yes** | 5 (1.0) | 2 (1.4) | 3 (1.1) | 0 (0.0) | .84* |
| **Received Other drugs to treat COVID-19: N (%) Yes** | 13 (2.5) | 6 (4.3) | 4 (1.4) | 3 (3.3) | .17* |

*Indicates results of an exact test

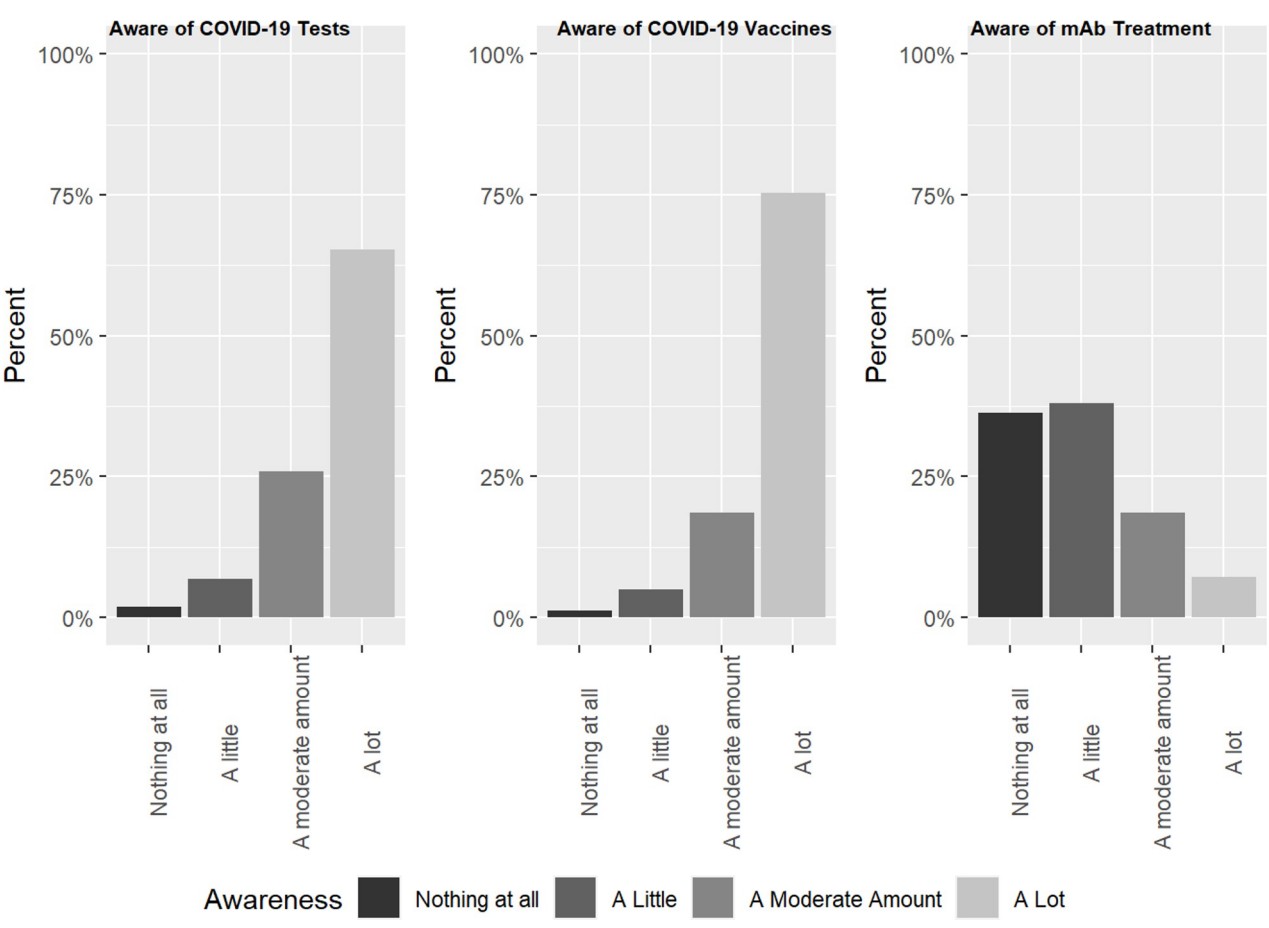

**Fig 1. Awareness of COVID-19 tests, vaccines, and mAb treatment in Colorado.**

to date;" [30]); only 5 (1%) respondents reported having received mAbs for COVID-19 while 13 (2.5%) reported receiving other medications to treat COVID-19. (Note: these data were collected before the availability of oral treatments for COVID-19.) There were no statistically significant racial/ethnic differences in testing, vaccination, or treatment for COVID-19, though this study was not designed with the intention to power to detect such differences. Among Hispanic/Latino survey respondents, compared to English-language survey respondents, Spanish-language survey respondents had significantly less awareness of COVID-19 vaccines (94.5% vs 83.4% moderate/a lot of awareness; $p = .026$) but greater awareness of mAbs (16.7% vs 38.9% moderate/a lot of awareness; $p = .01$).

Table 3 shows trust in monoclonal antibody treatment for COVID-19 overall and by racial/ethnic group for a variety of potential concerns (safety, efficacy, side effects, i.e., 'won't make someone sick', availability, affordability). A substantial proportion of respondents (~35–45%) indicated they didn't know enough to tell if they would trust mAbs or not. Among respondents who indicated some level of trust, the majority said they trusted at least somewhat that mAbs were safe and effective; trust was somewhat lower in availability and affordability. There were significant differences in trust in mAb safety and efficacy among racial/ethnic groups: White Non-Hispanic survey respondents were consistently highest in trust in safety, keeping someone out of the hospital, and not making someone sick (i.e., side effects). There were no significant racial/ethnic differences for mAb availability or affordability and no differences in trust between Hispanic/Latino English- vs Spanish-language survey respondents.

**Table 3. Trust in monoclonal antibody treatment for COVID-19 by Race/Ethnicity.**

| | Overall | Hispanic/ Latino/a | White Non-Hispanic | POC Non-Hispanic | p |
|---|---|---|---|---|---|
| **n** | 515 | 141 | 282 | 90 | |
| **Trust that mAbs are safe: N (%)** | | | | | .002* |
| Don't know | 206 (40.0) | 65 (46.1) | 97 (34.4) | 43 (47.8) | |
| Trust not at all/a little | 83 (16.1) | 25 (17.7) | 39 (13.8) | 19 (21.1) | |
| Trust some/ a lot | 226 (43.9) | 51 (36.2) | 146 (51.8) | 28 (31.1) | |
| **Trust that mAbs will work to keep someone out of the hospital: N (%)** | | | | | < .001* |
| Don't know | 178 (34.8) | 60 (42.9) | 85 (30.2) | 31 (36.0) | |
| Trust not at all/a little | 105 (20.5) | 29 (20.7) | 48 (17.1) | 28 (31.5) | |
| Trust some/a lot | 229 (44.7) | 51 (36.4) | 148 (52.7) | 29 (32.6) | |
| **Trust that mAbs won't make someone sick: N (%)** | | | | | .025* |
| Don't know | 220 (43.1) | 66 (47.5) | 114 (40.9) | 39 (43.3) | |
| Trust not at all/a little | 121 (23.7) | 35 (25.2) | 57 (20.4) | 29 (32.2) | |
| Trust some/a lot | 169 (33.1) | 38 (27.3) | 108 (38.7) | 22 (24.4) | |
| **Trust that mAbs: will be available: N (%)** | | | | | .15* |
| Don't know | 213 (41.6) | 60 (42.9) | 116 (41.4) | 36 (40.0) | |
| Trust not at all/a little | 171 (33.4) | 36 (25.7) | 102 (36.4) | 32 (35.6) | |
| Trust some/a lot | 128 (25.0) | 44 (31.4) | 62 (22.1) | 22 (24.4) | |
| **Trust that mAbs will be affordable: N (%)** | | | | | .80* |
| Don't know | 228 (44.5) | 65 (46.4) | 123 (43.9) | 39 (43.3) | |
| Trust not at all/a little | 184 (35.9) | 45 (32.1) | 106 (37.9) | 32 (35.6) | |
| Trust some/a lot | 100 (19.5) | 30 (21.4) | 51 (18.2) | 19 (21.1) | |

*Indicates results of an exact test

Table 4 shows willingness to receive mAbs oneself (among 391 respondents with at least one high-risk condition) and to recommend mAbs for a friend, family member, or another person as a proxy decision maker. Nearly 40% of respondents with 1+ high-risk condition indicated they were very worried about being hospitalized if they were to be diagnosed with COVID-19; Hispanic/Latino respondents indicated being much more worried about hospitalization (57.0%) than White Non-Hispanic (31.1%) and Other racial/ethnic groups (35.0%) respondents. Nearly all (97.7% yes/maybe) respondents reported they would consider getting mAb treatment if diagnosed with COVID-19 if it were recommended by a doctor; there were no significant differences in willingness to accept mAb treatment across racial/ethnic groups. Upon learning mAbs were delivered by IV infusion (note the survey was designed and launched before availability of subcutaneous injection options), about 16% indicated they were less willing to get the treatment; Hispanic/Latino and Non-Hispanic POC respondents were significantly less inclined to accept mAb treatment through IV infusion relative to White Non-Hispanic respondents. Respondents were generally willing to receive mAb treatment in a hospital (95.5% yes/maybe) or a specialty care center (93.8% yes/maybe); there was greater hesitation about receiving treatment in a mobile care unit (71.2% yes/maybe). There were significant racial/ethnic differences in willingness to receive mAb treatment across the various settings; while trends were similar to the overall findings, White Non-Hispanic respondents were generally less hesitant about getting treatment in any given setting than were Hispanic/Latino or Non-Hispanic POC respondents. Compared to English-language respondents, Spanish-language respondents reported more worry about hospitalization for both themselves (40.9% vs 86.8% very worried, p < .001) and others (64.8% vs 94.3% very worried, p = .002) and a greater likelihood of recommending mAb treatment to a family member or friend (42.6% vs 78.8%

**Table 4. Willingness to receive monoclonal antibody treatment for COVID-19 for Self and Others by Race/Ethnicity.**

| | Overall | Hispanic/ Latino/a | White Non-Hispanic | Non-Hispanic POC | P* |
|---|---|---|---|---|---|
| N (%) | 391 | 101 (25.8%) | 228 (58.3%) | 61 (15.6%) | |
| **Willingness—mAbs (self): N (%)** | | | | | .13* |
| No | 9 (2.3) | 2 (2.0) | 3 (1.3) | 4 (6.6) | |
| Yes | 251 (64.2) | 61 (60.4) | 154 (67.5) | 36 (59.0) | |
| Maybe | 131 (33.5) | 38 (37.6) | 71 (31.1) | 21 (34.4) | |
| **Worry about hospitalization (self): N (%)** | | | | | < .001* |
| Not at all worried | 31 (8.0) | 4 (4.0) | 18 (7.9) | 9 (15.0) | |
| A little/Somewhat worried | 208 (53.5) | 39 (39.0) | 139 (61.0) | 30 (50.0) | |
| Very worried | 150 (38.6) | 57 (57.0) | 71 (31.1) | 21 (35.0) | |
| **IV Infusion Influence Decision to receive mAbs: N (%)** | | | | | .004* |
| Less willing | 60 (15.9) | 22 (22.7) | 26 (11.7) | 12 (21.1) | |
| No difference | 292 (77.5) | 63 (64.9) | 185 (83.3) | 43 (75.4) | |
| More willing | 25 (6.6) | 12 (12.4) | 11 (5.0) | 2 (3.5) | |
| **Willingness—Infusion sites—Hospital: N (%)** | | | | | .031* |
| No | 18 (4.7) | 8 (8.3) | 5 (2.2) | 5 (68.5) | |
| Yes | 323 (85.2) | 75 (78.1) | 197 (88.3) | 50 (84.7) | |
| Maybe | 38 (10.0) | 13 (13.5) | 21 (9.4) | 4 (6.8) | |
| **Willingness—Infusion sites—Specialty Care Center: N (%)** | | | | | .051* |
| No | 23 (6.2) | 5 (5.6) | 11 (4.9) | 7 (11.9) | |
| Yes | 284 (76.1) | 61 (68.5) | 177 (79.0) | 45 (76.3) | |
| Maybe | 66 (17.7) | 23 (25.8) | 36 (16.1) | 7 (11.9) | |
| **Willingness—Infusion sites—Mobile Care Unit: N (%)** | | | | | < .001* |
| No | 105 (28.8) | 34 (39.5) | 47 (21.4) | 24 (41.4) | |
| Yes | 130 (35.6) | 21 (24.4) | 86 (39.1) | 22 (39.7) | |
| Maybe | 130 (35.6) | 31 (36.0) | 87 (39.5) | 11 (19.0) | |
| **Willingness–mAbs (others): N (%)** | | | | | .18* |
| No | 6 (2.2) | 2 (2.2) | 2 (1.6) | 2 (3.6) | |
| Maybe | 73 (26.9) | 63 (70.8) | 96 (76.2) | 21 (37.5) | |
| Yes | 192 (70.8) | 24 (27.0) | 28 (22.2) | 33 (58.9) | |
| **Recommend mAb to Family/Friend: N (%)** | | | | | .089* |
| No | 7 (2.7) | 3 (3.5) | 2 (1.7) | 2 (3.8) | |
| Maybe | 98 (37.7) | 48 (55.8) | 81 (67.5) | 26 (49.1) | |
| Yes | 155 (59.6) | 35 (40.7) | 37 (30.8) | 25 (47.2) | |

*Indicates result of an exact test

yes, p = .003). No other statistically significant differences by language of survey completion were found.

## Qualitative results

We completed eight focus groups with a total of 69 participants. Focus groups lasted an average of 70 minutes and had between 7 and 12 participants. Four of the six English-language focus groups were with urban/suburban participants and 2 with rural participants. The two Spanish-language focus groups had a mix of participants from urban/suburban and rural areas though the participants predominantly resided in urban/suburban areas. Similar to survey respondents, focus groups findings showed most focus group participants knew very little about mAbs. (Table 5, Quotes 1–2). Those who had heard something about mAbs most

**Table 5. Illustrative quotes for qualitative analysis of focus group themes.**

<u>Limited mABs knowledge overall</u>

**Quote 1**

*I saw [mAB] information and I sent it to my sister-in-law who had Covid. I didn't really know what it was.* Spanish participant #11

**Quote 2**

*"I have heard of it. Really, the only thing I know for sure is that it's supposed to be given when a person first has symptoms. Anything else about it, how much it costs or where you get it or how it's available, I have no idea."* English/rural participant #9

**Quote 3**

*Yeah. I just saw an article maybe a week ago or more that—I think it was about the governor of Texas was advocating for those treatments to be used. That was his major push. . .I'll echo what <name> and others have said, that I didn't—the first thing I heard about it was former president Trump getting it.* English/urban participant #3

**Quote 4**

*In my case, I've already had Covid last year. I had many symptoms like headaches, fever, and my bones ache a lot. If at that moment, somebody had told me about the antibodies and they had been an option for me, to feel better, maybe I would have done it. I was feeling really bad.*
*My husband is diabetic, he has high blood pressure, he's overweight, he was feeling worse than I was. Someone from our family was hospitalized for almost four months. If this option had been there last year, it could have been helpful for many people so that they wouldn't have gotten hospitalized or intubated.-*Spanish participant #19

**Quote 5**

*I had a brother who got COVID, and he's in Arkansas, and he was prescribed ivermectin, so we were talking about, why wasn't he given the opportunity for monoclonal antibodies versus ivermectin treatments? I know a little bit about it because we've talked about it because it affected us personally in our family.* English/rural participant #5

**Many, but not all, participants expressing willingness to get the treatment**

**Quote 6**

Yes, I would take it, and I would like every love of mine take it as well. I would like, if this is available, that it's not available just to certain groups of people. It should be available to everybody. Even if you have been vaccinated, you never know the reaction you have by contracting COVID, so this could be a lifesaver. English/urban participant #29

**Quote 7**

*I think I would consider using it. If I'm going to feel better and it's going to give me an option to not get hospitalized or get intubated, things like that; the way things are at the moment, we don't know how our bodies are going to react. Thank God, it's been a year since it happened to us, and we haven't gotten it again.* Spanish participant #19

**Quote 8**

*I don't really agree with [mAB treatment]. I have my own perception on why not, and I guess I would be considered a conspiracy theorist because I don't really appreciate manmade chemicals going into my body. . .it's nice that they're making something that's supposed to boost our immune system, but it's—if we all just took the time and ate right and ate broccoli and did our normal human things, we wouldn't be having to take something that was made in a lab to boost our immune system.*
English/rural participant #10

**Key Questions about mAB treatment**

*How do the benefits and side effects compare*?

**Quote 9**

*I would wanna know how is it made in the lab? How long does it stay in your body? How long would it be protective or not—or therapeutic. . . If there are side effects or other considerations that would need to be made.* English/urban participant #17

**Quote 10**

*Would the treatment be equally effective or nearly as effective on the different variants that are coming out?* English/rural participant #7

**Quote 11**

*"My only concern would be how it interacts with other drugs, and I'm sure the doctors would explain that to us before—in seeking our consent to have this."* English/urban participant #25

*Who is eligible to receive mABs and how can eligible people access the treatment?*

**Quote 12**

*I find it very interesting because you prevent it from reaching a more serious stage; you prevent hospitalization and other kinds of possible effects. I wonder why it's only for people who have grave conditions like being overweight and this kind of thing and not for any patient."* Spanish participant #6

**Quote 13**

*The first thing that I think of is the hoops that people have to jump through. One, you have to qualify with all of these other predisposed conditions. . .Then also, calling your doctor. . . getting to a referral coordinator to get the referral placed, to get the insurance to accept it. . .I don't mean to sound cynical, but it does seem like there are quite a few hoops than what it sounds like at face value.* English/urban participant #5

**Quote 14**

*I also see it as a bit complicated. When I go to make a medical appointment, they give me an appointment in three weeks. The idea of calling and getting an appointment fast and then asking the doctor to recommend this, the doctor has to recommend it and then, I don't know if—besides it's complicated in only 10 days.* Spanish participant #6

**Quote 15**

*"I believe that, for the average working citizen that has access to healthcare benefits, that it would be very beneficial for them. . .I just see it being a problem with the marginalized populations.*
English/rural participant #1

**Quote 16**

*"The timing, the cost, the logistics, but the transportation is a real critical issue too, especially for seniors and people who don't drive. They're not gonna give you service if you're positive for COVID. If you are getting sick, you're not going to feel like driving yourself or even attempting to get out of bed from what I hear from the symptoms. Yeah. How are they gonna get there? How are they gonna get the medication?"* English/urban participant #27

**Quote 17**

*I want to say that that is an inaccessible model to have infusion centers. Anybody that's disabled, homebound, or in congregate living can't access that. . .* English/rural participant #3

**Quote 18**

*I think the mobile bus thing sounds amazing. . . I don't see any reason why anybody in my community wouldn't take this if it was recommended by their physician.* English/rural participant #13

**Quote 19**

*Yes. Timing is everything. I think timing is the big issue. Where is the bus gonna be when? I would use a bus—if the timing was there.* English/rural participant #9

*How much does mAB treatment cost?*

**Quote 20**

*I'd wanna know, what would that actually cost, for myself and then for family members who are either not insured or underinsured–how much it would actually be to receive the treatment and what kinda medical bills they would look.* English/urban participant

**Quote 21**

*Yeah. I think cost is very important because, like they said, they cover the antibody, but they don't cover—our insurances aren't gonna cover the labs, or the government isn't gonna cover the fees that we're gonna get from the labs, and they can be very expensive.* English/urban participant #23

**Quote 22**

*Do I just go and do it and see if I get a surprise bill for $10,000 from an infusion center? I just had an infusion, and that's what it cost just for—walking in the door. I guess those pieces really concern me, that there's a lot of unknowns where I'm like, okay, the service itself is free, but everything else might end up putting me in serious debt, even though I'm trying to save my own life.* English/urban participant #26

**Quote 23**

*The service is free? But, in order to make an appointment with the doctor, the person needs to have, well. . . insurance. . .but for someone who doesn't have the monetary means, how can they access this service?* Spanish participant #1

*(Continued)*

**Table 5.** (Continued)

**Key Priorities for maB education and treatment**

*Spread the word*

**Quote 24**

*I consider that using the same means they used when the pandemic started, when we started to use the masks, 6 feet apart, washing our hands. When it started, we were bombarded; at the bus station, on the radio, on tv. I mean, if this treatment can save lives and can save lots of money in intensive care, then bombard it to make it massive. Because if it's only limited to information sites such as hospitals and that, there are people who never go.* Spanish participant #4

**Quote 25**

*Similar to what <name> mentioned, definitely, social media. I feel that's a big one personally for family and friends where they see news channels that share certain articles. Also, I feel like word of mouth is really big and often not considered. I feel it has to come from really key community leaders, trusted folks in the community that can share more about this information.* English/urban participant #4

**Quote 26**

*All of the things to everyone everywhere because it's so important and many people hear things through different channels. It's important to also—has been noted already to market and reach communities in their native languages where possible.* English/Urban Participant #17

**Quote 27**

*I was gonna say, we see how so much about misinformation and disinformation spreads on Facebook, so I think we have to counter the disinformation with the real information. It seems like Facebook, like you're gonna reach more people that way, even though I hate Facebook.* English/rural participant #5

**Quote 28**

*Around here, I'm in the [name of rural region], the small-town pages, like [name of Facebook page] My town's so small, we don't even have a page. The buy and sell pages and anywhere where the administration will let you. This is a community where people still hang up paper posters on storefronts if you are fortunate enough to have a storefront in your town.* English/rural participant #3

**Quote 29**

*I think that with everything that's been said; information should be transmitted via the web; at public places, hospitals, schools. It's very important that schools have this information, both, teachers and students to take home.* Spanish participant #3

*Make it Easy*

**Quote 30**

*Maybe this is a drastic step, but I would scrap the primary care referral thing. I think that that's too complicated, especially for really poor people.* English/urban participant #3

**Quote 31**

*Well, to make it a bit more accessible; something that didn't require a doctor's referral to be made. . . how are we going to receive a doctor's order, if you have COVID symptoms and you can't go to the clinic?* Spanish participant #18

***Quote 32***

*I also heard that they were doing drive-up sites in Florida and other states. I remember they said that instead of an infusion, they're giving people four injections, like one in each arm, one in the belly, and then one in the backside supposedly to get it in their system and have it the same potency of an infusion, but that way, they could do it in drive-up format, and that that that's how they were doing it in Florida and stuff like that.* English/urban participant

*Don't Make it Political*

**Quote 33**

*I was just going to say that I think because the COVID issue has been so politicized, that people are just going to have their own thoughts about whether or not they think, a certain champion, they can believe. It just makes it harder for folks to trust. . . the politicians seem to make a divide on it, between one party or the other, that just makes it harder.* English/urban participant #36

**Quote 34**

*The CDC has been awful at messaging. Whoever's the marketing and messaging, they need to go work for the CDC 'cause the CDC, basically, they say something, and then they alter it.* English/urban participant #29

**Quote 35**

*Yeah. I guess the question is, if you advertised, and everybody thought that this was great. . . do you think it would be a deterrent for somebody to get a vaccine? I guess that's one thing.* English/urban participant #2

**Quote 36**

*I was thinking, for my community in southwest Colorado, we have a lot of vaccine hesitancy, and I don't know how effective this would be for that kind of community where they don't want— let alone, they don't want to take the vaccine. Now, if they get COVID, then they will hesitate to try this treatment. I worry about that.* English/rural participant #14

referenced learning about it on the news after a politician or other famous person had received the treatment. (Quotes 3) One focus group participant had received the treatment and believed it had been of help. Several participants expressed dismay that mAbs had not been offered to a family member when ill with COVID-19 and suggested that lack of general knowledge may have contributed to the low use of mAbs. (Quotes 4,5) Knowledge about mAbs across English language focus groups was similar and, in contrast to survey findings, less among participants in Spanish-language focus groups. After learning more about mAbs through the resources shared during the session, participants generally had a favorable impression of mAbs. They expressed willingness to get it as it could lessen the severity of the disease and potentially prevent mortality. (Quotes 6,7). Participants were encouraged that this treatment offered hope despite limited treatment options. Some participants expressed skepticism and were unsure if they would be willing to receive mAbs if ill; a few stated they would decline this treatment if

offered. (Quote 8) It was more common among participants in rural and Spanish-language focus groups to have reduced willingness to consider or receive mAbs.

Focus group participants identified key questions related to mAbs to include in education or information about the treatment option: 1) How do the benefits and side effects compare?; 2) Who is eligible to receive mAbs, and how can eligible people access the treatment?; 3) How much does mAb treatment cost? Concerning benefits and side effects, participants had questions about the effectiveness of the treatment in general and across different variants, the durability of protection, side effects, interactions with other medications and health conditions and how this treatment differed from other treatments such as plasma infusions or antivirals. (Quotes 9–11). Participants recognized that answers to some of these questions might be particular to the individual. They expressed they would expect to have this conversation with a medical provider and wondered if medical providers were prepared to have these discussions. There were no notable differences in the type of information about benefits and risks across focus group participants' racial/ethnic or geographic characteristics.

Participants reported the need for clarity about high-risk populations eligible or prioritized for mAb treatment. Some participants wondered why supply could not be increased to allow access for all who might have COVID-19. (Quote 12) Many participants expressed doubt that people would be able to navigate the logistics to access treatment, especially because of the short timeframe between symptom onset and when treatment needed to be administered. (Quote 13) Participants worried they would not be able to access an appointment to obtain a treatment referral soon enough to then schedule treatment at an infusion center. (Quote 14) Discussions about who might be eligible and the process to access treatment prompted discussions of reaching the populations who might be most at risk of complications from COVID-19. Participants had concerns about equitable access by racial/ethnic minority, medically fragile, low-income, immigrant, uninsured, and unhoused populations. (Quotes 14–16) Concerns about equitable access stemmed from the multi-step process to obtain mAbs that required access to medical care for diagnosis and referral, reliable transportation, and navigating a complex healthcare system where most information is available only in English. All focus groups expressed concern about the limited number of infusion centers outside of the Denver Metro areas and were pleased to learn of mobile treatment centers that had recently become available. (Quote 18) Rural participants requested more specific information on the expected location of mobile treatment options. They expressed concern about accessing mobile treatment sites given long distances between incorporated areas, especially given the timeline for obtaining treatment might not coincide with when the mobile treatment site was close to their location. Spanish-language focus group discussions focused more on overcoming potential barriers to treatment access within the Latino immigrant community compared to English language focus groups' broader discussions of potential access disparities for multiple populations.

Costs associated with mAbs were of significant concern to focus group participants. Participants expressed that though the medication was free, there were other potential costs that could present a barrier to access for themselves or others. (Quotes 20, 21) Participants noted that insurance coverage for infusion costs could be hard to predict and that this could be an added stress to an already stressful situation when one was ill. (Quote 22) Discussions of cost also prompted discussion of equity as many people might not be able to afford the costs associated with mAb treatment and thus would not pursue it. (Quote 23) Cost discussions across focus group types were similar though Spanish-language focus groups did not dwell on this topic for long; participants acknowledged it as a significant barrier and then moved on to other topics.

Three priorities for mAb education and treatment emerged from the focus groups: 1) Spread the Word ("all of the things to everyone everywhere"); 2) Make it Easy; and 3) Don't

Make It Political. Based on the limited knowledge of participants and mAbs and their perception that others in their community also had limited knowledge, focus group participants emphasized the importance of spreading the word broadly across multiple media and via community leaders. (Quotes 24, 25) Participants acknowledged that there was not going to be one information source that would be a trusted source for a particular person, and thus use of multiple media and methods for disseminating information about mAbs was needed. (Quote 26) Sources of information that were trustworthy varied across focus groups. Urban/suburban focus group participants more often referenced the use of national sources like the Centers for Disease Control or reading scientific manuscripts directly than other groups. Many participants expressed that a recommendation from a healthcare provider would positively impact their decision to receive mAbs. A few English-speaking participants expressed mistrust of medical providers; Spanish-language participants did not. Participants acknowledged that misinformation was everywhere, particularly on social media and that it would be challenging to overcome the power of misinformation without a major campaign. (Quote 27) Rural participants emphasized local sources of information more than urban/suburban participants. (Quote 28) Spanish-language focus groups provide additional potential methods to disseminate mAb information that was particularly relevant to their community, including consular offices and flyers in the children's school folders. (Quote 29).

The priority to make it easy to obtain mAb treatment primarily focused on eliminating the requirement to get a provider referral for treatment. (Quotes 30–31) All focus group types felt this was critical to promoting equity and timely access to mAbs. Focus group participants also felt that more options for obtaining mAb treatment were needed, and some wondered if using injections rather than an infusion would expand access. (Quote 32) There were no differences across focus groups concerning the theme of "making it easy" to access mAbs. The theme, "Don't make it political," stemmed from concerns that information about the COVID-19 pandemic had been so polarizing. (Quote 33)Participants reported that they did not believe mAbs was favored by one political party affiliation vs another. Participants hoped the information about mAbs could be disseminated in a way acceptable across a broad range of political and social affiliations. Participants acknowledged that this would present a challenge as missteps in communication throughout the pandemic were a barrier to disseminating clear, actionable and neutral information about mAbs. (Quote 34). Participants also acknowledged that information about mAbs could complicate COVID-19 vaccine messaging, (Quote 35) with some wondering if vaccine hesitancy would translate to hesitancy to receive this treatment. (Quote 36) There was no notable difference between English-speaking urban and rural participants related to this theme. Spanish-language focus group participants, however, emphasized the lack of clarity in messaging related to the COVID-19 vaccine compared with English-language focus group participants in both urban and rural regions. Several Spanish-language focus group participants detailed negative experiences with the vaccine and hoped this treatment might provide an alternative to the "COVID vaccine people fear so much."

## Discussion

This study assessed community-level factors underlying equitable access to mAb treatment for COVID-19. Unlike COVID-19 vaccines and testing–which enjoyed broad awareness, if not acceptability–community members were largely unaware of the existence of effective COVID-19 treatment or when or how to access mAb treatment. This finding emerged from both the survey and focus group data. Lack of awareness of effective treatments for COVID-19 may contribute to unnecessary COVID-19 mortality and morbidity and persistent disparities in COVID-19 impact on racial/ethnic minority communities. Despite generally positive attitudes

toward mAbs and near-universal willingness to consider receiving mAb treatment, people expressed concern about the costs and logistics of accessing mAb treatment. These results suggested that a dissemination strategy should focus on promoting broad awareness of the availability of safe and effective treatment for those at high risk, with straightforward messaging on how to access treatment and any costs. Dissemination would also benefit from messaging tailored to specific community perspectives and systems of communication and influence–i.e., "all of the things to everyone everywhere," and addressing differences in trust, availability, and perceived effectiveness. Given that mAbs are most effective within a few days of experiencing symptoms, the messaging would also need to convey the need for timely testing to diagnose COVID-19. Though treatments for COVID-19 have evolved and mAb treatments are no longer the only evidence-based outpatient treatment available, our findings still provide important lessons for equitable dissemination and access to treatments for COVID-19. Applying these lessons to emerging treatments for COVID-19 may help address persistent disparities related to COVID-19.

Results showed that community members thought the messaging should be shared in many locations and through multiple media based on varied media preferences by age, race/ethnicity and political affiliation. Focus group participants endorsed public communications campaigns despite acknowledging that messaging throughout the COVID-19 pandemic was problematic. Participants also supported locally oriented and tailored messaging utilizing trusted community media, leaders, and healthcare providers. Participants recognized the challenges of implementing local messaging. For some, healthcare providers were a trusted source, but there was concern about healthcare providers having adequate knowledge of the treatment and the logistics of receiving it. Thus, a dissemination strategy is also needed to promote awareness among health care providers so they can be informed and proactive about educating and referring patients for treatment. Focus group participants shared a key concern about the challenges of implementing messaging at the local level and in a more personalized manner. From an equity perspective, community members expressed concern for those who lacked a trusted regular care provider or health insurance. Participants saw the role of other community leaders and organizations in spreading the word. They expressed the importance of having information on the treatment and how to access it connected to the COVID-19 testing process.

In addition to information dissemination, our findings support decreasing the logistical complexity of receiving mAb treatment. Focus group participants endorsed eliminating a required provider referral for treatment based on the time, costs and access barriers. This insight supported state-level policy eliminating the need for a referral, such as through a standing order for treatment [31]. While eliminating the provider referral could promote equity in the use of mAbs, it is plausible that it could widen disparities if mAb treatment access information is not equitably disseminated. Focus group participants supported broad dissemination but also emphasized that information campaign investments may need to be more targeted to communities that historically have less access to information and information that they feel applies to them. This suggests that dissemination must be designed to address current inequitable information access. Expanding access to treatment through treatment availability outside of "brick and mortar" infusion centers was highlighted as another strategy to decrease the logistical burden of mAb treatment. During focus groups, there was broad support for strategies like mobile treatment buses. However, survey respondents that were non-White had decreased willingness to receive mAb treatments in mobile care units. The reasons for this decreased willingness are unclear particularly concerning equitable access to and use of mAb treatment. The costs associated with mAb treatment was a key concern and may be quite complex to address in the US healthcare system. Focus group participants reported many ways people seeking mAb treatment could incur costs, from seeking care to obtaining the referral to

bills for service from the infusion center. Based on their prior experiences with health care, they had little trust that the treatment would be affordable and worried about the financial impact on themselves and less financially secure community members.

Across the methods used in this study, trust was a key line of inquiry. Mistrust in the healthcare system has been identified as a factor in healthcare disparities, and increased trust has been associated with improved healthcare outcomes [32–34]. Compared to Hispanic/Latino and Non-Hispanic POC respondents, White Non-Hispanic respondents consistently reported greater trust in mAb safety and efficacy but, surprisingly, no differences in trust in availability or affordability. There were also no statistically significant racial/ethnic differences among survey respondents in their willingness to receive mAb treatment, although there were racial/ethnic differences in willingness to receive IV treatment in hospitals, specialty care centers, or mobile treatment sites. During focus groups, we noted comparatively less willingness to receive mAb treatment by Spanish-language and rural focus group participants. These participants expressed skepticism about the chemical components of the mAb treatment, side effects, and effectiveness of treatments. The skepticism expressed in Spanish-language focus groups may partly explain the data demonstrating that the state of Colorado had some of the widest disparities in COVID-19 vaccination between Latinos and Whites compared with other states [35]. In contrast, Spanish-language survey respondents expressed more trust in healthcare providers to provide a treatment recommendation. They were more likely to recommended mAbs to a family member or friend than English-language Hispanic language survey respondents. These findings likely indicate that multiple factors influence trust and that the impact of trust on health disparities among Latinos generally and among subgroups of the Latino population remains misunderstood. A better understanding of how trust intersects with the COVID-19 disparities for Latinos could provide key insights into general health equity for this population. Finally, the frequent changes in messaging—necessitated by fluctuations in our understanding of the disease, treatment evidence, and accessible care—were seen as detrimental to earning community trust in public health communications.

The rapid evolution of evidence and changing circumstances are unfortunate but unavoidable pandemic phenomena, and communicating uncertainty is challenging, but mistrust in public health communication may have longstanding impacts. Despite focus group participant enthusiasm for broad dissemination campaigns, psychological research demonstrates that negative experiences often have a greater impact on subsequent behavior and emotions than positive ones. Thus the negativity bias towards COVID-19 communications may be difficult to overcome [36].

## Limitations

The denominator for the number of people eligible to participate and who were exposed to survey recruitment materials is unknown, and thus we cannot report a survey response rate. Survey respondents self-reported being high-risk for COVID-19; this was not independently verified. Despite making the survey available in multiple languages, recruitment efforts did not reach those who spoke languages other than English or Spanish (except for one Amharic-speaking respondent). While partnerships with community organizations facilitated the representation of participants who were Hispanic/Latino at a rate higher than state demographics, we lacked partnerships with other important groups who also experience COVID-19 disparities, such as refugee communities from the Middle East, Africa, and Asia [37]. Thus, this study did not reveal their perspectives on mAb treatment for COVID-19. We also could not compare differences within other discrete racial groups represented in the broader POC category due to the small sample size. Finally, we did not undertake member checking for qualitative findings

for logistical reasons; thus, the focus group insights may be missing some community perspectives.

## Conclusion

Our findings indicate a critical need for better dissemination of general information about mAb treatment, especially among marginalized communities. Given the low levels of knowledge about mAbs we found across all survey demographics, there is a need for information dissemination to "everyone, everywhere". At the same time, given glaring disparities in COVID-19 infection, morbidity and mortality, the design of communication strategies must be rooted in promoting equity, which means making information available in varied community locations, multiple languages, appropriate for populations with varying degrees of literacy, and with clear guidance for those facing barriers to healthcare access. As the pandemic progresses, mAbs are no longer the first line treatment [38], but these priorities for information dissemination remain relevant. Equitable information dissemination alone, however, will not be sufficient to promote equity in access and use of treatments for COVID-19. Removing barriers related to who can prescribe or refer to treatment and increasing options for where people can access treatment will also play a key role. Finally, these findings underscore the need for broad stakeholder buy-in for generating solutions to public health issues in ways that are tailored to individual community perspectives, assets, and systems of communication and influence. For instance, involving community members and organizations that have a direct path to bring messages back to their constituents in a culturally responsive manner could be a way to build the infrastructure of information dissemination that would improve equity outcomes in the future.

## Supporting information

**S1 Appendix.** S1A File. Community Survey–English. S1B File. Community Survey–Spanish. (ZIP)

**S2 Appendix.** S2A File. Focus Group Guide–English. S2B File. Focus Group Guide–Spanish. (ZIP)

## Acknowledgments

We would like to thank and acknowledge the important contributions of the community members and partners on the mAb Colorado project's stakeholder advisory panel. These partners were critical in informing the design, conduct, recruitment, interpretation, and dissemination of findings from this research. We thank Ms. Jenn Jones, Ms. Anowara Begum, and Ms. Madelaine Carter for their assistance with conducting this research and/or preparing this manuscript for submission. We acknowledge the important contributions of Colorado's regional health connectors and community research liaisons for supporting recruitment in diverse communities. We also thank the participants in this research, without whom this work would not be possible.

## Author Contributions

**Conceptualization:** Bethany M. Kwan, Chelsea Sobczak, Matthew K. Wynia, Adit A. Ginde, Lisa Ross DeCamp.

**Data curation:** Chelsea Sobczak, Carol Gorman, Samantha Roberts.

**Formal analysis:** Chelsea Sobczak, Carol Gorman, Samantha Roberts, Lisa Ross DeCamp.

**Funding acquisition:** Bethany M. Kwan, Matthew K. Wynia, Adit A. Ginde.

**Investigation:** Bethany M. Kwan, Carol Gorman, Vanessa Owen, Lisa Ross DeCamp.

**Methodology:** Bethany M. Kwan, Samantha Roberts, Matthew K. Wynia, Adit A. Ginde, Lisa Ross DeCamp.

**Project administration:** Bethany M. Kwan, Chelsea Sobczak, Carol Gorman, Vanessa Owen, Griselda Pena-Jackson, Lisa Ross DeCamp.

**Resources:** Griselda Pena-Jackson.

**Supervision:** Bethany M. Kwan, Adit A. Ginde, Lisa Ross DeCamp.

**Visualization:** Bethany M. Kwan, Chelsea Sobczak, Carol Gorman, Samantha Roberts, Matthew K. Wynia, Adit A. Ginde, Owen Ziegler, Lisa Ross DeCamp.

**Writing – original draft:** Bethany M. Kwan, Chelsea Sobczak, Carol Gorman, Samantha Roberts, Lisa Ross DeCamp.

**Writing – review & editing:** Bethany M. Kwan, Chelsea Sobczak, Carol Gorman, Samantha Roberts, Vanessa Owen, Matthew K. Wynia, Adit A. Ginde, Griselda Pena-Jackson, Owen Ziegler, Lisa Ross DeCamp.

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
