## [Decision Letter · Decision Letter 0]

15 Sep 2022

PONE-D-22-23297“All of the things to everyone everywhere”: A mixed methods analysis of community perspectives on equitable access to monoclonal antibody treatment for COVID-19PLOS ONE

Dear Dr. Kwan,

Thank you for submitting your manuscript to PLOS ONE. After careful consideration, we feel that it has merit but does not fully meet PLOS ONE’s publication criteria as it currently stands. Therefore, we invite you to submit a revised version of the manuscript that addresses the points raised during the review process.

We look forward to receiving your revised manuscript.

Kind regards,

Kehinde Kazeem Kanmodi, BDS

Academic Editor

PLOS ONE

“No authors have competing interests”

Reviewers' comments:

Reviewer's Responses to Questions

**Comments to the Author**

1. Is the manuscript technically sound, and do the data support the conclusions?

Reviewer #1: Yes

Reviewer #2: Yes

Reviewer #3: Partly

2. Has the statistical analysis been performed appropriately and rigorously? 

Reviewer #1: Yes

Reviewer #2: Yes

Reviewer #3: No

3. Have the authors made all data underlying the findings in their manuscript fully available?

Reviewer #1: Yes

Reviewer #2: Yes

Reviewer #3: Yes

4. Is the manuscript presented in an intelligible fashion and written in standard English?

Reviewer #1: Yes

Reviewer #2: Yes

Reviewer #3: Yes

5. Review Comments to the Author

Reviewer #1: This is a well-written manuscript (if a little long) and I enjoyed reading it.

The following issues should be addressed in order to make the ms suitable for publication:

Abstract:

• The Background section needs a further sentence or two, clarifying the rationale for the study. Why was it important to elicit peoples’ perspectives on this topic?

Introduction:

• P. 6: “For instance, the state of Colorado set up a secure web-based system was set up in concert with the state health department to enable community health care providers to refer patients for mAb treatment at one of the multiple infusion sites across the state.” – This sentence contains a grammatical error – please revise.

• P. 7: “PCORNet” – please define upon first use.

• P. 7: the paragraph beginning with “Thus, concerns about the inequitable access to and use of mAbs among marginalized patients with COVID-19 appear legitimate and observable nationally.” This paragraph needs to present a stronger rationale for eliciting community perspectives, as opposed to provider perspectives (which would help highlight problems with logistics etc.). You mention community-level knowledge, cultural, or psychosocial factors which could play a role. It is worth going into more detail here. Is there prior research in related areas which has shown access problems due to such underlying issues? It would be useful, for example, to draw on research on COVID-19 vaccination and ethnic minorities here.

Method:

• Participants and Recruitment: where was the target number of 450 survey participants derived from? I see no results of a prospective power analysis reported. Please clarify.

• Please provide some information about how the community stakeholder advisory panel was formed – this is mentioned a couple of times in the Method section, but little detail is provided on it.

Results:

• Qualitative Results: what was the average size of a focus group? What was the average duration?

• P.21 last line: “It is was” – please correct.

• Table 5: participants should be numbered, or pseudonyms used.

Discussion:

• There needs to be an acknowledgement and discussion of the fact that increased vulnerability towards COVID-19 was reliant upon participants’ self-report. Lack of member-checking and other methods to ensure trustworthiness of your results also needs to be discussed.

Reviewer #2: Good piece of work. However, a justification for your choices would be useful for the reader. Please justify your rationale for your specific choices highlighted in your manuscript.

Also, in your qualitative results section, indicate quotes by each statement you make. This makes it easy for the reader to relate. Put in the quotes table at the end of the qualitative results section.

Reviewer #3: May I start by saying Congratulations to the authors for coming up with this robust work.

However, there are a few comments note worthy which if addressed will improve the quality of your publication.

1. The 1st sentence of the last paragraph of the introduction stated what the study did, which is very correct. It's important to note that it's is always better to clearly stated the purpose or aim of a study.

Subsequent sentences in the paragraph looked like a description of some of the process in the methods.

Authors should kindly find a way to incooporate this in the method section.

2. It also looks like there is a repetition in the descriptions in the last part of the introduction and the 1st paragraph of the methods section.

Authors should kindly address it

3. Please what informed the authors decision to target 450 participants in the survey?

4. How was the minimum Sample size calculated, especially for the quantitative aspect?

Please kindly note that PlosOne is a visible journal and when eventually your work is published, other researchers may want to make reference to your work and methods to formulate theirs.

Authors should kindly be more explicit on smoke size calculation.

5. Reference 7 was not well written. Kindly look out for the article information and get the requisite particulars of the article.

6. Reference 17 was not well written. Kindly look at the recommended NLM citation style for the article by statpearls publishing

7. Reference is was not well written. No year of publication and edition of publication.

8. Reference 20 was not well written too. Granted that some particulars are not shown in the publication. Authors should at least ensure URL or DOI is included

6. PLOS authors have the option to publish the peer review history of their article (what does this mean?). If published, this will include your full peer review and any attached files.

Reviewer #1: **Yes: **Judith Eberhardt

Reviewer #2: No

Reviewer #3: **Yes: **Ugochukwu Anthony Eze

Please I want my name to appear but do not think it's appropriate to let the comment come public

---

## [Author Response · Author response to Decision Letter 0]

13 Oct 2022

Comments from Reviewer 1

Comment Response

1) Abstract: The Background section needs a further sentence or two, clarifying the rationale for the study. Why was it important to elicit peoples’ perspectives on this topic? We have added the following sentence to the abstract: “Reasons for underuse and inequity may include community member lack of awareness or healthcare access barriers, among others.”

2) Introduction: P.6 “For instance, the state of Colorado set up a secure web-based system was set up in concert with the state health department to enable community health care providers to refer patients for mAb treatment at one of the multiple infusion sites across the state.” – This sentence contains a grammatical error – please revise. This sentence has been edited. Thank you for pointing this out.

3) Introduction: P.7 “PCORNet” – please define upon first use. This was corrected

4) Introduction: P. 7 The paragraph beginning with “Thus, concerns about the inequitable access to and use of mAbs among marginalized patients with COVID-19 appear legitimate and observable nationally.” This paragraph needs to present a stronger rationale for eliciting community perspectives, as opposed to provider perspectives (which would help highlight problems with logistics etc.). You mention community-level knowledge, cultural, or psychosocial factors which could play a role. It is worth going into more detail here. Is there prior research in related areas which has shown access problems due to such underlying issues? It would be useful, for example, to draw on research on COVID-19 vaccination and ethnic minorities here. Thank you for this comment. There are several earlier paragraphs in the introduction that go into depth on the multiple factors underlying use of mABs for COVID-19 in general, including the need to understand both patient and provider perspectives. The paragraph noted in this comment specifically points out the need to assess factors that may contribute to disparities. To enhance the rationale for the key point in this paragraph (health disparities can be attributed to both psychosocial and logistical factors), we have added a reference to Derose et al’s paper in Health Affairs on individual and system-level factors underlying health disparities and access. On page 7, we state, “A public health perspective on health disparities considers both individual-level factors such as health beliefs and system-level factors such as strategies to ensure the availability of care [18].” 

5) Methods: Participants and Recruitment: where was the target number of 450 survey participants derived from? I see no results of a prospective power analysis reported. Please clarify. We have added a statement about the sample size estimation on pages 8-9. 

6) Methods: Please provide some information about how the community stakeholder advisory panel was formed – this is mentioned a couple of times in the Method section, but little detail is provided on it. We have added a description of the SAP on page 8 as follows, “The SAP was comprised of 25 individuals including 12 community members, 3 healthcare providers, 2 public health department representatives, and 8 regional health connectors (RHCs). SAP members were recruited through professional contacts and existing relationships with community organizations and practice-based research networks. The RHCs are a community-based workforce in Colorado based in health organizations across the state with the goal of identifying and addressing health issues with their regions.”

7) Results: what was the average size of a focus group? What was the average duration? We have now included this information on page 22.

8) Results: last line: P.21 “It is was” – please correct. This was corrected

9) Results: Table 5: participants should be numbered, or pseudonyms used. We have now identified participants by number in Table 5.

10) Discussion: There needs to be an acknowledgement and discussion of the fact that increased vulnerability towards COVID-19 was reliant upon participants’ self-report. Lack of member-checking and other methods to ensure trustworthiness of your results also needs to be discussed. We have added these two points to the limitations. 

Comments from Reviewer 2

Comment Response

1) Good piece of work. However, a justification for your choices would be useful for the reader. Please justify your rationale for your specific choices highlighted in your manuscript.

Also, in your qualitative results section, indicate quotes by each statement you make. This makes it easy for the reader to relate. Put in the quotes table at the end of the qualitative results section. We utilized a table for quotes for ease of reading the manuscript as a whole. We believe having the quotes in the text and a table would be redundant. We defer to the editors if they would like the quotes in the manuscript text and in the table.

2) Methods: P. 12 “Any identifying information was removed from the transcript. Transcripts were not returned to participants for comment or correction.” What was the rationale? It was not feasible to return transcripts to participants as their participation in the project was limited to their time during the focus group. This clarification has been added to the text. 

3) Methods: P. 12 “We performed sub-analyses to compare responses among Hispanic participants by the language of survey completion (Spanish vs. English). We created a bar plot to show all respondents’ awareness of COVID-19 tests, vaccines, and treatments.” What was the rationale? The rationale for subanalyses by Spanish/English was to determine whether there were additional concerns about mAb access among those with a language barrier, beyond cultural differences. We have added this statement to the methods on page 13. It’s not clear that a rationale is needed for the decision to use a bar plot to present findings. 

4) Methods: P. 13 “We did not complete member-checking with participants, but findings were shared with the project’s community stakeholder advisory panel, who endorsed findings that reflected community member experiences and viewpoints.” What was your rational? A justification would be useful.

We have now added justification to this sentence. 

5) Results: P. 21, 26-29 Quotes 6-16, 22-36 I would indicate the quote for easy read and still maintain the table below See response to similar comment above.

6) Results: Table 5: I would indicate the table at the end of the qualitative results. It’s not clear what this comment means. 

Comments from Reviewer 3

Comment Response

1) May I start by saying congratulations

to the authors for coming up with this robust work. However, there are a few comments noteworthy which if addressed will improve the quality of your publication. The 1st sentence of the last paragraph of the introduction stated what the study did, which is very correct. It's important to note that it’s always better to clearly stated the purpose or aim of a study.

Subsequent sentences in the paragraph looked like a description of some of the process in the methods.

Authors should kindly find a way to incorporate this in the method section. Thank you for your kind words. 

We have integrated the final paragraph of the introduction into the method section as requested.

2) It also looks like there is a repetition in the descriptions in the last part of the introduction and the 1st paragraph of the methods section.

Authors should kindly address it We have addressed this as in the comment above.

3) Please what informed the authors decision to target 450 participants in the survey? How was the minimum Sample size calculated, especially for the quantitative aspect?

Please kindly note that PlosOne is a visible journal and when eventually your work is published, other researchers may want to make reference to your work and methods to formulate theirs.

Authors should kindly be more explicit on smoke size calculation. We have addressed this in response to a similar comment from Reviewer 1.

4) Reference 7 was not well written. Kindly look out for the article information and get the requisite particulars of the article. We have edited the citation for Ref 7.

5) Reference 17 was not well written. Kindly look at the recommended NLM citation style for the article by statpearls publishing This has been corrected

6) Reference is was not well written. No year of publication and edition of publication. This comment does not specify which reference is meant here. We can make additional edits to the references cited during a copy editing phase if needed.

7) Reference 20 was not well written too. Granted that some particulars are not shown in the publication. Authors should at least ensure URL or DOI is included This has been corrected

---

## [Decision Letter · Decision Letter 1]

27 Oct 2022

“All of the things to everyone everywhere”: A mixed methods analysis of community perspectives on equitable access to monoclonal antibody treatment for COVID-19

PONE-D-22-23297R1

Dear Dr. Bethany M. Kwan,

We’re pleased to inform you that your manuscript has been judged scientifically suitable for publication and will be formally accepted for publication once it meets all outstanding technical requirements.

Kind regards,

Kehinde Kazeem Kanmodi, BDS

Academic Editor

PLOS ONE

Additional Editor Comments (optional):

Nil.

Reviewers' comments:

Reviewer's Responses to Questions

**Comments to the Author**

1. If the authors have adequately addressed your comments raised in a previous round of review and you feel that this manuscript is now acceptable for publication, you may indicate that here to bypass the “Comments to the Author” section, enter your conflict of interest statement in the “Confidential to Editor” section, and submit your "Accept" recommendation.

Reviewer #1: All comments have been addressed

Reviewer #3: (No Response)

2. Is the manuscript technically sound, and do the data support the conclusions?

Reviewer #1: Yes

Reviewer #3: Yes

3. Has the statistical analysis been performed appropriately and rigorously? 

Reviewer #1: Yes

Reviewer #3: Yes

4. Have the authors made all data underlying the findings in their manuscript fully available?

Reviewer #1: Yes

Reviewer #3: Yes

5. Is the manuscript presented in an intelligible fashion and written in standard English?

Reviewer #1: Yes

Reviewer #3: Yes

6. Review Comments to the Author

Reviewer #1: My comments appear to have been addressed satisfactorily. I now recommend acceptance and publication.

Reviewer #3: I have carefully reviewed and the manuscript and made some explicit comments and recommend for modification in the areas of methods and references. I have also read through the responses to my comments and they have been adequately addressed by the author.

The work is quite topical and lends a voice to address the COVID-19 vaccine inequity the at exist between populations.

The article is of global health importance.

7. PLOS authors have the option to publish the peer review history of their article (what does this mean?). If published, this will include your full peer review and any attached files.

Reviewer #1: **Yes: **Dr Judith Eberhardt

Reviewer #3: **Yes: **Dr. Ugochukwu Anthony Eze

---

## [Editor Report · Acceptance letter]

14 Nov 2022

PONE-D-22-23297R1 

“All of the things to everyone everywhere”: A mixed methods analysis of community perspectives on equitable access to monoclonal antibody treatment for COVID-19 

Dear Dr. Kwan:

I'm pleased to inform you that your manuscript has been deemed suitable for publication in PLOS ONE. Congratulations! Your manuscript is now with our production department. 

Kind regards, 

on behalf of

Dr. Kehinde Kazeem Kanmodi 

Academic Editor

PLOS ONE